# A computational reproducibility study of PLOS ONE articles featuring longitudinal data analyses

**Heidi Seibold**[1,2,3,4]*, **Severin Czerny**[1], **Siona Decke**[1], **Roman Dieterle**[1], **Thomas Eder**[1], **Steffen Fohr**[1], **Nico Hahn**[1], **Rabea Hartmann**[1], **Christoph Heindl**[1], **Philipp Kopper**[1], **Dario Lepke**[1], **Verena Loidl**[1], **Maximilian Mandl**[1], **Sarah Musiol**[1], **Jessica Peter**[1], **Alexander Piehler**[1], **Elio Rojas**[1], **Stefanie Schmid**[1], **Hannah Schmidt**[1], **Melissa Schmoll**[1], **Lennart Schneider**[1], **Xiao-Yin To**[1], **Viet Tran**[1], **Antje Völker**[1], **Moritz Wagner**[1], **Joshua Wagner**[1], **Maria Waize**[1], **Hannah Wecker**[1], **Rui Yang**[1], **Simone Zellner**[1], **Malte Nalenz**[1]

**1** Department of Statistics, LMU Munich, Munich, Germany, **2** Data Science Group, University of Bielefeld, Bielefeld, Germany, **3** Helmholtz AI, Helmholtz Zentrum München, Munich, Germany, **4** LMU Open Science Center, LMU Munich, Munich, Germany

* heidi@seibold.co

**Data Availability Statement:** All results including detailed reports and code for each of the 11 papers are available in the GitLab repository https://gitlab.com/HeidiSeibold/reproducibility-study-plos-one.

## Abstract

Computational reproducibility is a corner stone for sound and credible research. Especially in complex statistical analyses—such as the analysis of longitudinal data—reproducing results is far from simple, especially if no source code is available. In this work we aimed to reproduce analyses of longitudinal data of 11 articles published in PLOS ONE. Inclusion criteria were the availability of data and author consent. We investigated the types of methods and software used and whether we were able to reproduce the data analysis using open source software. Most articles provided overview tables and simple visualisations. Generalised Estimating Equations (GEEs) were the most popular statistical models among the selected articles. Only one article used open source software and only one published part of the analysis code. Replication was difficult in most cases and required reverse engineering of results or contacting the authors. For three articles we were not able to reproduce the results, for another two only parts of them. For all but two articles we had to contact the authors to be able to reproduce the results. Our main learning is that reproducing papers is difficult if no code is supplied and leads to a high burden for those conducting the reproductions. Open data policies in journals are good, but to truly boost reproducibility we suggest adding open code policies.

## Introduction

Reproducibility is—or should be—an integral part of science. While computational reproducibility is only one part of the story, it is an important one. Studies on computational reproducibility (e.g. [1–6]) have found reproducing findings in papers is far from simple. Obstacles

All files can also be accessed through the Open Science Framework (https://osf.io/xqknz).

**Funding:** This research has been supported by the German Federal Ministry of Education and Research (BMBF) under Grant No. 01IS18036A (Munich Center of Machine Learning) to HS.

**Competing interests:** The authors have declared that no competing interests exist.

include lack of methods descriptions and no availability of source code or even data. Researchers can choose from a multitude of analysis strategies and if they are not sufficiently described, the likelihood of being able to reproduce the results are low [7, 8]. Even in cases where results can be reproduced, it is often *tedious and time-consuming* to do so [6].

We conducted a reproducibility study based on articles published in the journal PLOS ONE to learn about reporting practices in longitudinal data analyses. All PLOS ONE papers which fulfilled our selection criteria (see Fig 1) in April 2019 were chosen ([9–19]).

Longitudinal data is data containing repeated observations or measurements of the objects of study over time. For example, consider a study investigating the effect of alcohol and marijuana use of college students on their academic performance [10]. Students perform a monthly survey on their alcohol and marijuana use and consent to obtain their grade point averages (GPAs) each semester during the study period. In this study not only the outcome of interest (GPAs during several semesters) is longitudinal, but also the covariates (alcohol and marijuana use) change over time. This does not always have to be the case in longitudinal data analysis. Covariates may also be constant over time (e.g. sex) or baseline values (e.g. alcohol consumption during the month before enrollment).

Due to the clustered nature of longitudinal data with several observations per subject, special statistical methods are required. Common statistical models for longitudinal data are mixed effect models or generalized estimating equations. These models can have complex structures and rigorous reporting is required for reproducing model outputs. A study on reporting in generalized linear mixed effect models (GLMMs) on papers from 2000 to 2012 found that there is room for improvement on reporting of these models [20]. Alongside the models, visualization of the data often plays an important role in analyzing longitudinal data. An example is the spaghetti plot, a line graph with the outcome on the y-axis and time on the x-axis. Research on computational reproducibility when methods are complex—such as in this case—is still in its infancy. With this study we aim to add to this field and to provide some

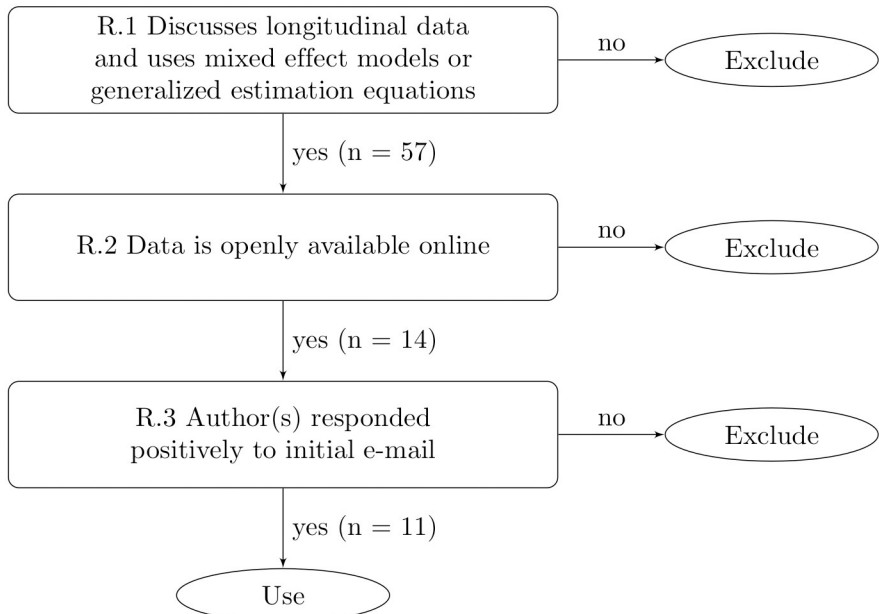

**Fig 1. Data selection.** Data selection procedure according to our requirements and number of papers fulfilling the respective requirements.

insights on challenges of reproducibility in the 11 papers investigated. Furthermore we would like to note that each reproduced paper, is another paper that we can put more trust in. As such reproducing a single paper is already a relevant addition to science.

Computational (or analytic [21]) reproducibility studies—as we define them for this work—take existing papers and corresponding data sets and aim to obtain the same results from the statistical analyses. One prerequisite for such a study is the access to the data set which was used for the original analyses. Also, a clear description of the methods used is essential. An easily reproducible paper provides openly licensed data alongside an openly licensed source code in a programming language commonly used for statistical analyses and also available under a free open source software license (e.g. R [22] or python [23]). If the source code is accompanied with a detailed description of the computing environment (e.g. operating system and versions of R packages) or the computing environment itself (e.g. a Docker container [24]) we believe the chances of obtaining the exact same results to be highest. It is difficult to determine whether a scientific project is *reproducible*: Is it possible to obtain exactly the same values? Is the (relative) deviation lower than a certain value? Is the difference in p-value lower than a certain value? These and more are questions that can be asked and if answered "yes" the results can be marked as reproducible. Yet all of these come with downsides including being too strict, incomparable, uncomputable, or downright not interesting. Here, we use the definition of leading to the same interpretation, without a rigorous formal definition. The reason is, that the papers analysed here use very different models, so it is hard to compare them on a single scale (such as absolute relative deviation, see e.g. [6]). We argue, that in combination with a qualitative description of challenges and difficulties that we faced in each reproduction process, this definition fits our small scale, heterogeneous, setting better.

In this work we investigated longitudinal data analyses published in PLOS ONE. The multidisciplinarity of PLOS ONE is a benefit for our study as longitudinal data play a role in various fields. Additionally the requirement for a data availability statement in PLOS ONE (see https://journals.plos.org/plosone/s/data-availability) facilitates the endeavour of a reproducibility study. Note that we only selected papers which provided data openly online and where authors agreed with being included in this study. We assume that this leads to a positive bias in the sense that other papers would be more difficult to reproduce.

In the following we discuss the questions we asked in this reproducibility study, the setup of the study within the context of a university course, the procedure of paper selection, and describe the process of reproducing the results.

## Materials and methods

### Study questions

The aim of this study is to investigate reproducibility in a sample of 11 PLOS ONE papers dealing with longitudinal data. We also collect information on usage of methods, how they are made available and computing environments used. We expect that this study will help future authors in making their work reproducible, even in complex settings such as when working with longitudinal data. Note that based on the selection of 11 papers we cannot make inferences on papers in general or in the journal. We can, however, learn from the obstacles we encountered in the given papers. Also, even reproducing a single paper creates scientific value. It provides a scientific check of the work and increases (or in case of failure decreases) trust in the results.

With the reproducibility study we want to answer the following questions:

1. Which methods are used?

(a) What types of tables are shown?

(b) What types of figures are shown?

(c) What types of statistical models are used?

2. Which software is used?

(a) Is the software free and open source?

(b) Is the source code available?

(c) Is the computing environment described (or delivered)?

3. Are we able to reproduce the data analysis?

(a) Are the methods used clearly documented in the paper or supplementary material (e.g. analysis code)?

(b) Do we have to contact the authors in order to reproduce the analysis? If so, are authors responsive and helpful? How many e-mails are needed to reproduce the results?

(c) Do we receive the same (or very similar) numbers in tables, figures and models?

4. What are characteristics of papers which make reproducibility easy/possible or difficult/ impossible?

5. What are learnings from this study? What recommendations can we give future authors for describing their methods and reporting their results?

## Project circumstances

This project was conducted as part of the master level course *Analysis of Longitudinal Data* running during the summer term 2019 (23.01.19—27.07.19) at the Ludwig-Maximilians-Universität München. The course is a 6 ECTS (credit points according to the European Credit Transfer and Accumulation System) course aimed at statistics master students (compulsory in biostatistics master, elective in other statistics masters) with 4 hours of class each week: 3 hours with a professor (Heidi Seibold), 1 with a teaching assistant (Malte Nalenz). The course teaches how to work with longitudinal data and discusses appropriate models, such as mixed effect models and generalized estimating equations, and how to apply them in different scenarios. As part of this course, student groups (2-3 students) were assigned a paper for which they aimed to reproduce the analysis of longitudinal data. In practical sessions the students received help with programming related problems and understanding the general theory of longitudinal data analysis. To limit the likelihood of bias due to differing skills of students, all groups received support from the teachers. Students were advised to contact the authors directly in case of unclear specifications of methods. Internal peer reviews, where one group of students checked the setup of all other groups, ensured that all groups had the same solid technical and organizational setup. Finally all projects were carefully evaluated by the teachers and updated in case of problems. Replications and a student paper were the output of the course for each student group and handed in in August 2019. We believe that the setup of this reproducibility study benefits from the large time commitment the students put into reproducing the papers. Also having several students and two researchers work on each paper, ensures a high quality of the study.

This project involved secondary analyses of existing data sets. We had not worked with the data sets in question before.

## Selection of papers

For a paper to be eligible for the reproducibility study it has to fulfill the following requirements:

**R.1** The paper deals with longitudinal data and uses mixed effect models or generalized estimating equations for analysis.

**R.2** The paper is accompanied by data. This data is freely available online without registration.

**R.3** At least one author is responsive to e-mails.

Requirement **R.1** allows us to select only papers relevant to the topic of this project. Requirement **R.2** is necessary to allow for reproducing results without burdens (e.g. application for data access). Although PLOS ONE does have an open data policy (https://journals. plos.org/plosone/s/data-availability), we found many articles which had statements such as "Data cannot be made publicly available due to ethical and legal restrictions". Issues with data policies in journals have been studied in [25]. Requirement **R.3** is important to be able to contact the authors later on in case of questions. Fig 1 shows the selection procedure. All papers which did not fulfill the criteria were excluded. The PLOS website search function was utilized to scan through PLOS ONE published works. Key words used were "mixed model", "generalized estimating equations", "longitudinal study" and "cohort study". This key word search—performed for us by a contact at PLOS ONE—resulted in 57 papers. From these 14 papers fulfilled all criteria and were selected. Two authors prohibited to use of their work within our study. We note that authors do not have the right to prohibit the reuse of their work as all papers are published under CC-BY license. However the negative response lead us to drop the papers, as we expected to have the need to contact authors with questions. For one paper we did not receive any response. Discussions on the selection criteria of all proposed papers are documented in https://osf.io/dx5mn/?branch=public.

Table 1 shows a summary of all papers selected so far.

## Replication

In the reproducibility study we adhered to open science best practices. (1) We contacted all corresponding authors of papers we aimed to reproduce via e-mail; (2) all of our source code and data used is available; (3) any potential errors in the original publications were reported immediately to the corresponding author.

In our study we conducted all analyses as close to the original analyses as possible. If many analyses were performed in the original paper, we focused on the analyses of longitudinal data. We conducted all analyses using R [22] regardless of the software used in the original paper to mimic a situation where no access to licensed software is available (R was the only open source software used in the 11 papers).

Each analyis consisted of the following steps:

1. Read the data into R.

2. Prepare data for analysis.

3. Produce overview figure(s) with outcome(s) on the y-axis and time on the x-axis.

4. Reproduce analysis results (e.g. model coefficients, tables, figures).

The description about all these steps was generally vague (see classification of reported results in [6]) meaning that there were multiple ways of preparing or analysing the data that were in line with the descriptions in the original paper. This study, thus, exposed a large

**Table 1. Selected papers.**

| | Citation | Title |
|---|---|---|
| [9] | Wagner et al (2017) | Airway Microbial Community Turnover Differs by BPD Severity in Ventilated Preterm Infants |
| [10] | Meda et al (2017) | Longitudinal Influence of Alcohol and Marijuana Use on Academic Performance in College Students |
| [11] | Visaya et al (2015) | Analysis of Binary Multivariate Longitudinal Data via 2-Dimensional Orbits: An Application to the Agincourt Health and Socio-Demographic Surveillance System in South Africa |
| [12] | Vo et al (2018) | Optimizing Community Screening for Tuberculosis: Spatial Analysis of Localized Case Finding from Door-to-Door Screening for TB in an Urban District of Ho Chi Minh City, Viet Nam |
| [13] | Aerenhouts et al (2015) | Estimating Body Composition in Adolescent Sprint Athletes: Comparison of Different Methods in a 3 Years Longitudinal Design |
| [14] | Tabatabai et al (2016) | Racial and Gender Disparities in Incidence of Lung and Bronchus Cancer in the United States: A Longitudinal Analysis |
| [15] | Rawson et al (2015) | Association of Functional Polymorphisms from Brain-Derived Neurotrophic Factor and Serotonin-Related Genes with Depressive Symptoms after a Medical Stressor in Older Adults |
| [16] | Kawaguchi, Desrochers (2018) | A Time-Lagged Effect of Conspecific Density on Habitat Selection by Snowshoe Hare |
| [17] | Lemley et al (2016) | Morphometry Predicts Early GFR Change in Primary Proteinuric Glomerulopathies: A Longitudinal Cohort Study Using Generalized Estimating Equations |
| [18] | Carmody et al (2018) | Fluctuations in Airway Bacterial Communities Associated with Clinical States and Disease Stages in Cystic Fibrosis |
| [19] | Villalonga-Olives et al (2017) | Longitudinal Changes in Health Related Quality of Life in Children with Migrant Backgrounds |

amount of "researcher degrees of freedom" [26] coupled with a lack in transparency about in the original studies. We aimed to take steps that align as closely as possible with the original paper and the results therein. That means, if the methods description in paper or supplementary material were clear, we used those; If not, we tried different possible strategies that we assumed could be correct; If this was not possible or did not lead to the expected results, we contacted the authors to ask for help. All code used by us is publicly available including software versions and in a format easily readable by humans (literate programming, for further information see section on technical details).

## Results

The results of our study are summarized in Tables 2–4. As each paper has its own story and reasons why it was or wasn't reproducible and what the barriers were, we provide a short description of each individual paper reproduction.

**Which methods are used?** For an overview on the following questions we refer to Table 2.

**What types of tables are shown?** Most of the papers show tables on characteristics of the observation units at baseline or other summary tables (similar to the so called "Table 1" commonly used in biomedical research) which give a good overview of the data.

**What types of figures are shown?** Few papers include classical visualizations taught in courses on longitudinal data, such as spaghetti plots. They mostly present other visualizations (for details, see Table 2).

**Table 2. Which statistical methods were used by the papers?.**

|      | Overview Tables | Visualisations | Models Used |
|------|-----------------|----------------|-------------|
| [9]  | Baseline demographics | Several, e.g. spaghetti plot | Beta Binomial Mixed Model |
| [10] | Baseline demographics, model output | Several, e.g. scatter plots (alkohol vs. marijuana use) of different time points | LMM |
| [11] | Overview of household types | Several, e.g. lasagna plot | GEE |
| [12] | Baseline demographics | none | GEE |
| [13] | Correlation | none | LMM (cross-classified) |
| [14] | Many especially smoking and lung cancer incidence rates for different year, genders, races and regions | Mean curves | LMM |
| [15] | Baseline demographics | Mean curves | GEE |
| [16] | Data overview | Mean curves | GEE |
| [17] | Correlation matrix | Mean curves | GEE |
| [18] | Sample characteristics | Several, e.g. FEV1 over time | GEE |
| [19] | Baseline demographics | DAG | GEE |

**What types of statistical models are used?** Although in most cases (G)LMMs are superior to GEEs (see [27] for an in-depth discussion and further references)—, 7 out of the 11 papers used GEEs for their analyses [11, 12, 15–19]. There is, in fact, only one complex mixed model among the methods used (Beta Binomial Mixed Model, [9]). The other articles [10, 13, 14] use LMMs which are equivalent to GEEs for normally distributed response variables. It should be noted that the selection of papers may not be representative of the general use of GEEs and (G)LMMs. Nevertheless it seems that the reluctance of using GEEs has not spilled over from the statistics community to some other fields, which we speculate to have historical reasons, as GLMMs used to be difficult to compute.

**Which software is used?** The results of this section are summarized in Table 3.

**Is the software free and open source?** All except one paper (paper [16]) used closed source software. As our goal was to evaluate how hard reproducing results is when licenses for software products are not available we worked with the open source software R. Implementations in different software products for complex methods such as GEEs and (G)LMMs may show slightly different results even when given the same inputs and with this we expected difficulties in reproducing exactly the same numbers for all papers using software other than R.

**Table 3. Which software was used by the papers?.**

|      | Software | Open Source | Source Code | Computing Environment |
|------|----------|-------------|-------------|-----------------------|
| [9]  | SAS | no | partly | SAS version |
| [10] | SPSS | no | no | SPSS version |
| [11] | no information (email contact states Stata) | no | no | no information |
| [12] | no information (email contact states Stata) | no | no | no information |
| [13] | SAS | no | no | SAS version |
| [14] | SAS | no | no | SAS version |
| [15] | SAS | no | no | SAS version |
| [16] | R | yes | upon request | Package version |
| [17] | SAS | no | no | SAS version |
| [18] | SPSS | no | no | SPSS version |
| [19] | MPlus | no | no | MPlus version |

**Table 4. Were the results reproducible?.**

| | Method documentation | Contact Attempts | Author Responses | Models Computable | Same Interpretation | Classification of Failure |
|---|---|---|---|---|---|---|
| [9] | Missing Details | 2 | 1 | partly | no | Software differences |
| [10] | Missing Details | 0 | 0 | yes | yes | |
| [11] | yes | 1 | 1 | partly | yes | Software differences |
| [12] | Missing Details | 1 | 1 | yes | yes | |
| [13] | Missing Details | 3 | 2 | partly | no | Software differences |
| [14] | yes | 1 | 0 | no | no | Software differences, Model Description |
| [15] | Correlation Structure missing | 1 | 1 | yes | yes | |
| [16] | Correlation Structure missing | 1 | 1 | yes | yes | |
| [17] | Correlation Structure missing | 3 | 1 | yes | yes | |
| [18] | | 4 | 1 | no | | Data and Model description |
| [19] | yes | 0 | 0 | yes | yes | |

**Is the source code available?**   Only one paper (paper [9]) provided source code. The source code provided was only a small part of the entire code needed to reproduce the results. Nevertheless it was a major help in obtaining the specifications of the models. For one paper we received the code through our email conversations [16]. For all other papers we had to rely on the methods and results sections of the papers. Often we resorted to reverse engineering the results as the methods sections were not sufficiently detailed.

**Is the computing environment described (or delivered)?**   In most cases the authors provided information on the software used and the software version (9 out of 11). None of the papers described the operating system or provided a computing environment (e.g. Docker container).

**Are we able to reproduce the data analysis?**   The results of this section are summarized in Table 4.

**Are the methods used clearly documented in paper or supplementary material (e.g. analysis code)?**   Although all papers in question had methods sections, for most papers we were not able to extract all needed information to reproduce the results by ourselves. The most common issue was that papers did not provide enough detail about the methods used (e.g. model type was mentioned but no detailed model specifications, for details see Table 4). Since, in addition, no source code was provided (except for paper [9]), reproducing results was generally only possible by reverse engineering and/or contacting the authors. As most authors used licensed software which was not available to us, we could not determine if we would have reached the same results using default settings in the respective software. A clear documentation therefore requires enough detail to explicitly specify all necessary parameters for the model, even when using a different software.

**Do we have to contact the authors in order to reproduce the analysis? How many e-mails are needed to reproduce the results?**   In all but two cases (papers [10, 19]) we contacted the authors to ask questions on how the results were generated (for four of them several emails were exchanged). All but one of the authors responded, which was to be expected as we had previously contacted them asking whether they would agree with us doing this project and only papers were chosen where authors responded positively. In most cases responses by authors were helpful.

**Do we receive the same (or very similar) numbers in tables, figures and models?**   As the articles use different models and present their main results in terms of different statistics (model coefficients, F-statistics, correlation), the purely numerical deviation between our

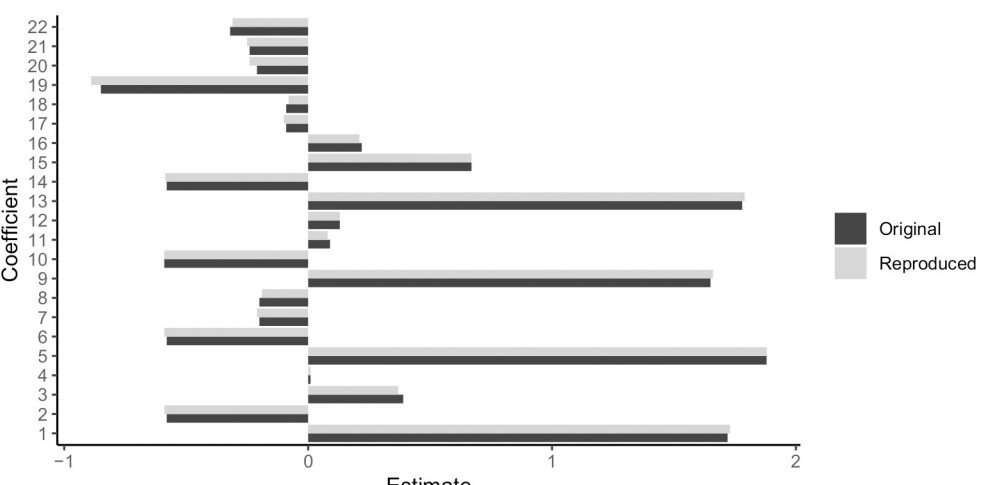

**Fig 2. Original and reproduced model parameter estimates for the ewbGEE model of article [15].** In this article the differences in parameters do not lead to a different interpretation.

results and the original results is not informative in isolation. Also, as we used different software implementations, some deviation was to be expected. Therefore, we define similar results as having the same implied interpretations, regarding sign and magnitude of effects. If the signs of the coefficients was the same and the ordering and magnitude of coefficients roughly the same, we regarded the results as successfully reproduced. We were able to fully reproduce 6 out of 11 articles (see also Table 4). Here differences were marginal and did not lead to a change of interpretations. An example (original and reproduced coefficients of article [15]) can be seen in Fig 2. For another two articles at least parts of the analysis could be reproduced (e.g. one out of two models used by the authors). For the 8 articles, that we found to be fully or partly reproducible, we were able to follow the data preprocessing and identify the most likely model specifications. Only three out of the 11 papers could not be reproduced at all, one because of implementation differences [13] and one due to problems preparing the data set used by the authors [18]. In [14] it was unclear how the data was originally analysed and without responses from the authors to our contact attempts via email we were not able to determine whether the different conclusions reached by our analysis are due to incorrect analysis on side of the authors or missing information.

Note that for some of the results, a considerable amount of time and effort needed to be invested to reverse engineer model settings. In the following we summarize the reproduction process for each paper individually, in order to give more insights about the specific problems and challenges that we encountered. (see also Table 4).

In [9] problems arose with the provided data set. The data description was found to be insufficient. Variable names in the data set differed from the ones in the code provided by the authors. We were able to resolve this problem based on feedback from the authors. When running the analysis using R and the R package PROreg [28], results differed from the original results due to details in the implementation and a different optimization procedure. The reproduced coefficients had the same sign as in the original study. However, differences in magnitude were large for some of the coefficients, likely due to differences in the optimization procedure. Given our definition, we were unable to reproduce the results. A second model fitted by the authors was not reproduced, due to convergence problems (model could not be fitted at all).

We were able to reproduce the results in [10] without contacting the authors. Some difficulty arose from the very sparse model description in the publication, such as, which variables were included as fixed or random effects. Also no source code was available. However within reasonable trial of different model specifications we obtained very similar results as in the original publication.

In [11] the number of observations differed between the publication and the provided data set. Upon request one of the authors provided a data set, that was almost identical to the one used in the study. The performed descriptive analysis and correlation analysis yielded the same results. A second difficulty arose, as the authors did not specify the correlation structure used in their model, but instead relied on the Stata routine to determine the best fitting correlation structure using the Quasi-Likelihood information criterion. If the correlation structure yielding the coefficients closest to the ones in [11] is used, the coefficients are almost identical. However, we also performed the aforementioned model search procedure in R but ended up with a different correlation structure as the best fitting. Using the correlation structure found best by our R implementation, would lead to a change in interpretation of the coefficients.

In [12] difficulties arose from different implementations in the software used. Also the model description was incomplete, which required us to try all possible combinations of variables to include. However, the correlation structure was well described and with feedback from the authors we were able to obtain the same results deviating only on the third decimal.

[13] used a cross-classified LMM, via the SAS "PROC mixed procedure". Reproduction in R was difficult, as no R package offered the exact same functionality. After trying several R implementations, we settled on the nlme R package [29]. The random effects were not specified in the publication. Also SAS code to shed light on this question was not available. Other questions regarding preprocessing and model specifications could be resolved through the feedback of the authors, but we did not receive the needed information on the random effects. As such we could not reproduce the results.

In [14] the data set used for modeling was not given as a file. Instead the authors provided links to the website where the data had been initially obtained from. We were not able to obtain the same data set given the sources and the description. This might be due to changes in the online sources. Still, differences in summary statistics were not substantial. We were unable to reproduce the same model due to unclear model specification. Our attempts led to some vastly different estimates. Possible reasons for failure are an insufficient model description or even incorrect analysis.

We were able to reproduce the results in [15] with only minor differences in the estimated coefficients. Feedback from the authors was required to find the correct correlation structure used in their GEE model, which was not explicitly stated in the paper.

The results in [16] were computationally reproducible. Despite minor differences in the coefficients we arrived at the same interpretations and differences were most likely due to different optimization procedures in the softwares used. The correlation structure was not stated in the article, but we were able to find the correct one using reverse engineering (grid search).

For the reproduction of [17] we had problems with data preprocessing. This was partly due to the unclear handling of missing values and due to details of the dimensionality reduction procedure used in preprocessing. The authors provided the final data set when we contacted them. The model specifications of the GEE used by the authors were not stated, but we were able to reproduce the exact same results as the authors by reverse engineering the correlation structure and link function. During this we found that using different model specifications or slightly different versions of the data set leads to substantially different results. Given the above definition this article was reproducible.

The results in [18] could not be reproduced. The (DNA) data was given in raw format as a collection of hundreds of individual files, without any provided code or step by step guide for preprocessing, making reproduction of the data set to be used in the statistical analysis impossible for us. Figures and Tables of the clinic data were reproducible.

The results in [19] were reproducible. All necessary model specifications for their GEE model and reasoning behind it were explicitly stated in the paper. The original analysis was carried out in M-plus, but reproduction in R gave almost identical results.

**What are characteristics of papers which make reproducibility easy/possible or difficult/impossible?**   Based on the discussion of the individual papers we identified determinants of successes and failures. We found that the simpler the methods used in the paper the easier it was to reproduce the paper. Papers dealing with classical LMMs (papers [10, 14]) were reasonably easy to reproduce.

The data provided by the authors played a major role as well. If the clean data was provided, reproducing was much easier than for papers providing raw data (papers [14, 17, 18]), where preprocessing was still necessary. For one paper [18] getting and preparing the data was so complex that we gave up. Even after the authors provided us with an online tutorial on working with this type of data, we were far from understanding what needed to be done. If specialists (e.g. bioinformaticians) on working with this type of data had been involved, we might have had better chances.

We believe that with code provided—even if it is written using software we do not have access to—computational reproducibility is easier to obtain. It is hard to make this conclusion based on the 11 papers we worked with, because only one provided partial code and 1 provided code on request, but they also did not contradict our prior beliefs.

**What are learnings from this study? What recommendations can we give future authors for describing their methods and reporting their results?**   Trying to reproduce 11 papers gave us a glimpse at how hard computational reproducibility is. We used papers published in an open access journal, which provided data and the authors were supportive of the project. We think it is fair to assume that these papers are among the most open projects available in academic literature at the moment. Nevertheless we were only able to reproduce the results without contacting the authors for two papers.

We not only recommend authors to provide data **and** code with their paper, but we suggest that this should be made a requirement from journals.

## Further points

One paper published raw names of study participants, which we saw as unnecessary information and with that as an unreasonable breach of the participants. We informed the authors who updated the data on the journal website.

## Discussion

In this study we aimed at reproducing the results from 11 PLOS ONE papers dealing with statistical methods for longitudinal data. We found that most authors use tables and figures as tools for presenting research results. Although all papers in question had data available for download, only one paper came with accompanied source code. From our point of view the lack of source code is the main barrier in reproducing results of the papers. For some papers we were still able to reproduce results by using a strategy of reverse engineering the results and by asking the authors. In an ideal situation, however, the information needed should not be hidden within the computers and minds of original authors, but should be shared as part of

the article (optimally in the form of a research compendium with paper, data, code, and metadata).

One of the authors initially contacted asked us to refrain from reproducing their paper on the grounds that students would not have the capabilities to do such complex analyses. We did not include the article in our study, but strongly disagree with this statement, especially since the students in question all have a strong statistics background and benefited from the guidance of researchers. Furthermore the students checked each other's works in an internal peer review. We would even go so far as to claim that a lot of other statistical work is less understood by the researcher and less thoroughly checked by peers before it is combined into a publication. Working as a big team gave us the option to conduct time intensive reverse engineering attempts of results, which small research teams or single researchers would potentially not have had.

We did not choose the papers randomly, but based on the set of potential papers given to us by PLOS ONE and then selected all papers meeting our criteria (see Fig 1). We can and should not draw conclusions from our findings on the 11 selected papers on the broader scientific landscape. Our work does, however, give us some insights on what researchers, reviewers, editors and publishers could focus on improving in the future: Publish code next to the data. To PLOS ONE we propose to include code in their open data policy.

Reproducing a scientific article is an important contribution to science and knowledge discovery. It increases trust in the research which is computationally reproducible and raises doubt in the research which is not.

## Technical details

All results including detailed reports and code for each of the 11 papers are available in the GitLab repository https://gitlab.com/HeidiSeibold/reproducibility-study-plos-one. All files can also be accessed through the Open Science Framework (https://osf.io/xqknz). For all computations all relevant computational information (R and package versions, operating system) are given below the respective computations. The relevant information for this article itself is shown below.

- R version 4.0.3 (2020-10-10), `x86_64-pc-linux-gnu`

- Locale: `LC_CTYPE=en_US.UTF-8`, `LC_NUMERIC=C`, `LC_TIME=de_DE.UTF-8`, `LC_COLLATE=en_US.UTF-8`, `LC_MONETARY=de_DE.UTF-8`, `LC_MESSAGES=en_US.UTF-8`, `LC_PAPER=de_DE.UTF-8`, `LC_NAME=C`, `LC_ADDRESS=C`, `LC_TELEPHONE=C`, `LC_MEASUREMENT=de_DE.UTF-8`, `LC_IDENTIFICATION=C`

- Running under: `Ubuntu 20.04.2 LTS`

- Matrix products: default

- BLAS: `/usr/lib/x86_64-linux-gnu/blas/libblas.so.3.9.0`

- LAPACK: `/usr/lib/x86_64-linux-gnu/lapack/liblapack.so.3.9.0`

- Base packages: base, datasets, graphics, grDevices, methods, stats, tools, utils

- Other packages: data.table 1.13.0, dplyr 1.0.2, ggplot2 3.3.3, googlesheets 0.3.0, kableExtra 1.3.1, knitr 1.32, plyr 1.8.6, rcrossref 1.1.0

- Loaded via a namespace (and not attached): cellranger 1.1.0, cli 2.4.0, codetools 0.2-18, colorspace 2.0-0, compiler 4.0.3, crayon 1.4.1, crul 1.1.0, curl 4.3, digest 0.6.27, DT 0.18, ellipsis 0.3.1, evaluate 0.14, fansi 0.4.2, farver 2.1.0, fastmap 1.1.0, generics 0.0.2, glue 1.4.2, grid

4.0.3, gtable 0.3.0, hms 0.5.3, htmltools 0.5.1.1, htmlwidgets 1.5.3, httpcode 0.3.0, httpuv 1.5.5, httr 1.4.2, jsonlite 1.7.2, labeling 0.4.2, later 1.1.0.1, lifecycle 1.0.0, magrittr 2.0.1, mime 0.10, miniUI 0.1.1.1, munsell 0.5.0, pillar 1.6.0, pkgconfig 2.0.3, promises 1.2.0.1, ps 1.6.0, purrr 0.3.4, R6 2.5.0, Rcpp 1.0.6, readr 1.4.0, reshape2 1.4.4, rlang 0.4.10, rmarkdown 2.7, rstudioapi 0.13, rvest 0.3.6, scales 1.1.1, shiny 1.6.0, stringi 1.5.3, stringr 1.4.0, tibble 3.1.1, tidyselect 1.1.0, utf8 1.2.1, vctrs 0.3.7, viridisLite 0.4.0, webshot 0.5.2, withr 2.4.2, xfun 0.22, xml2 1.3.2, xtables 1.8-4

## Author Contributions

**Conceptualization:** Heidi Seibold.

**Formal analysis:** Severin Czerny, Siona Decke, Roman Dieterle, Thomas Eder, Steffen Fohr, Nico Hahn, Rabea Hartmann, Christoph Heindl, Philipp Kopper, Dario Lepke, Verena Loidl, Maximilian Mandl, Sarah Musiol, Jessica Peter, Alexander Piehler, Elio Rojas, Stefanie Schmid, Hannah Schmidt, Melissa Schmoll, Lennart Schneider, Xiao-Yin To, Viet Tran, Antje Völker, Moritz Wagner, Joshua Wagner, Maria Waize, Hannah Wecker, Rui Yang, Simone Zellner.

**Investigation:** Heidi Seibold.

**Methodology:** Heidi Seibold.

**Project administration:** Heidi Seibold, Malte Nalenz.

**Software:** Heidi Seibold.

**Supervision:** Heidi Seibold, Malte Nalenz.

**Visualization:** Heidi Seibold, Malte Nalenz.

**Writing – original draft:** Heidi Seibold, Malte Nalenz.

**Writing – review & editing:** Heidi Seibold, Malte Nalenz.

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
