## [Decision Letter · Decision Letter 0]

6 Oct 2020

PONE-D-20-25993

A computational reproducibility study of PLOS ONE articles featuring longitudinal data analyses

PLOS ONE

Dear Dr. Seibold,

Thank you for submitting your manuscript to PLOS ONE. After careful consideration, we feel that it has merit but does not fully meet PLOS ONE’s publication criteria as it currently stands. Therefore, we invite you to submit a revised version of the manuscript that addresses the points raised during the review process.

The two reviewers and I agree that your study was well done, clearly reported, and relevant, and that your work contributes to our understanding of reproducibility of studies using longitudinal analyses. Yet, several issues need to be dealt with in your revision.

First, the two reviewers requested some additional clarity of the type of submission (registered report or not). I apologize for not indicating to the reviewers that your work was originally submitted as a registered report but later handled as a standard submission. Please clarify this. 

Second, both reviewers asked you to provide additional details of the methods, sampling, operationalizations, and data access. Both reviewers provided detailed feedback on the reporting and analyses that I ask you to consider as you revise your submission.

Third, the reviewers indicate that the main goals and main results could be presented more clearly in several sections of the manuscript. It is also import to be clear on your definition of reproducibility as you present the main results in the abstract. 

We look forward to receiving your revised manuscript.

Kind regards,

Jelte M. Wicherts

Academic Editor

PLOS ONE

Journal Requirements:

3.  In your manuscript text, we note that "We did not choose the papers randomly, but based on the set of potential papers given to us by PLOS ONE and then selected all papers meeting our criteria". In the Methods, please ensure that you have included the following:

- Information about how the initial set of 57 papers was selected, including any inclusion/exclusion criteria applied, so that other interested researchers can reproduce this analysis. We would also recommend that the complete set of 57 initial papers be provided as a supplementary information file.

- The complete inclusion/exclusion criteria used to select the initial set of 14 papers from the 57 papers that were identified.

- The complete inclusion/exclusion criteria used to select the final set of 11 papers.

4. Please amend the manuscript submission data (via Edit Submission) to include authors Severin Czerny,  Siona Decke, Roman Dieterle, Thomas Eder, Steffen Fohr, Nico Hahn, Rabea Hartmann, Christoph Heindl, Philipp Kopper, Dario Lepke, Verena Loidl, Maximilian Mandl, Sarah Musiol, Jessica Peter, Alexander Piehler, Elio Rojas, Stefanie Schmid, Hannah Schmidt, Melissa Schmoll, Lennart Schneider, Xiao-Yin To, Viet Tran, Antje Volker, Moritz Wagner, Joshua Wagner, Maria Waize, Hannah Wecker, Rui Yang, Simone Zellner.

Reviewers' comments:

Reviewer's Responses to Questions

**Comments to the Author**

1. Is the manuscript technically sound, and do the data support the conclusions?

Reviewer #1: Partly

Reviewer #2: Yes

2. Has the statistical analysis been performed appropriately and rigorously? 

Reviewer #1: Yes

Reviewer #2: I Don't Know

3. Have the authors made all data underlying the findings in their manuscript fully available?

Reviewer #1: No

Reviewer #2: Yes

4. Is the manuscript presented in an intelligible fashion and written in standard English?

Reviewer #1: Yes

Reviewer #2: Yes

5. Review Comments to the Author

Reviewer #1: Manuscript ID: PONE-D-20-25993

Manuscript title: A computational reproducibility study of PLOS ONE articles featuring longitudinal data analyses

Summary

This article reports a retrospective observational study designed to test the analytic reproducibility of a small set of PLOS ONE articles containing longitudinal analyses. Specifically, as a part of a class exercise, the authors and their students attempted to repeat the original analyses performed in the selected articles to see if they could obtain similar results. A range of difficulties reproducing the original results were encountered – some of which could be resolved through contact with the original authors. The generalizability of the results is quite limited due to the small sample size and somewhat ad-hoc sampling procedures; however, the authors appropriately calibrate their conclusions to the evidence, for example stating that “We can and should not draw conclusions from our findings on the 11 selected papers on the broader scientific landscape.”

Generally, the paper is clearly written and is concise, though I think some important information is absent (see detailed comments below). The study appears to be transparently reported with analysis scripts and data for each reproducibility attempt made publicly available in an Open Science Framework repository. I only checked this repository superficially – one issue is that I could not seem to identify a data file for the study (see comment below) which needs to be addressed.

Important note: After reading the paper and writing my review, I was surprised to find a document included after the reference section with the title “Review and response to the registered report”. I was not informed that this study was a registered report and as far as I can tell this is not mentioned in the manuscript aside from this appended document. Can the circumstances of this study be clarified? If this is a registered report then I feel I should be given more information. Most importantly, I need to see the original registered protocol. I would also like to see the prior review history and know whether the original stage 1 reviewers are also appraising the stage 2 manuscript.

Major Comments

- The exact operationalization of several of the outcome variables could be made clearer. For example, for the question “Are the methods used clearly documented in paper or supplementary material (e.g. analysis code)?” what was considered ‘clear’ vs. ‘unclear’? For “Do we receive the same (or very similar) numbers in tables, figures and models?” what was considered ‘similar’ vs. ‘dissimilar’? For “What are characteristics of papers which make reproducibility easy/possible or difficult/impossible?” – exactly what characteristics were examined?

- The oversight of the student work could be described in more detail – did teachers fully

- In the methods section, the sampling procedure is somewhat unclear – for example, “57 papers were initially screened. The PLOS website search function was utilized to scan through PLOS ONE published works. Key words used were “mixed model”, “generalized estimating equations”, “longitudinal study” and “cohort study”. 14 papers fulfilled the criteria and were selected”. Where does the number 57 come from? For the 14 papers – did they fulfill the search criteria? Or the criteria in Fig 1? Or both?

- It seems to me that the selection criteria will have heavily biased the results in favour of positive reproducibility outcomes – for example, only studies that had confirmed data availability were selected and only studies where authors replied to contact and were favourable to the reproducibility attempt were included. Because these factors probably influenced the results quite substantially, I’d suggest this bias is mentioned in key parts of the paper like the abstract and introduction.

- I examined the OSF repository (https://osf.io/xqknz/) for this study and it was unclear to me where to find the data for this study (I could find data for the studies that the authors attempted to reproduce). Could clear instructions be provided on how to find and access the study data?

- Could it be clarified if all analyses reported in eligible papers were examined or just the subset pertaining to longitudinal analyses?

- It is reported that for one article partial analysis code was available and for a second article the full analysis code was made available during email exchanges with the original authors. Its not clear whether all original authors were explicitly asked to make their code available – could this be clarified? If so, what were the responses to this query? Were reasonable justifications provided for not sharing the original code?

- The operational definition of reproducibility could be made clearer in the methods section (and perhaps also in the introduction) – in the results section the authors state “we define similar results as having the same implied interpretations” – this seems to be a less strict definition than used in other studies of analytic reproducibility (e.g., Hardwicke et al., 2018; 2020; Minocher et al., 2020). Some clarification and comment on this would be helpful for understanding the results.

- I think it would be informative to mention in the abstract how many analyses were reproducible only when assistance was provided by original authors.

- This sentence in the discussion is unclear – “We did not choose the papers randomly, but based on the set of potential papers given to us by PLOS ONE and then selected all papers meeting our criteria (see Figure 1).” If the papers were given to the authors by PLOS ONE then this needs to be mentioned and explained at least in the methods section.

Minor Comments

- Terminology usage in this domain is diverse and sometimes contradictory (see e.g., https://doi.org/10.1126/scitranslmed.aaf5027) – I’d recommend including explicit definitions and avoiding use of loaded terminology if possible. For example, it would be good to have a clear definition of ‘computational reproducibility’ in the opening paragraph. The authors may also want to consider using the term ‘analytic reproducibility’ instead of computational reproducibility. Researchers in this domain have recently started to draw a distinction between the two concepts and the former seems more applicable to what the present study has addressed. The distinction is discussed in this article - https://doi.org/10.31222/osf.io/h35wt – specifically, “Computational reproducibility is often assessed by attempting to re-run original computational code and can therefore fail if original code is unavailable or non-functioning (e.g., Stodden et al., 2018; Obels et al., 2019). By contrast, analytic reproducibility is assessed by attempting to repeat the original analysis procedures, which can involve implementing those procedures in new code if necessary (e.g., Hardwicke et al., 2018; Minocher et al., 2020)”

- An additional point on terminology - use of the term ‘replication’ (e.g., in the abstract and introduction) should perhaps be avoided if possible in this context because it is often used to mean “repeating original study methods and obtaining new data” – whereas here it is being used synonymously with computational reproducibility to mean “repeating original study analyses on the original data” (see http://arxiv.org/abs/1802.03311)

- I felt the study design could be made much more explicit in the introduction. For example, “The articles we chose are [1–11]” – briefly mentioning the sampling procedure would be helpful here so the reader can understand the study design (e.g., was it a random sample, arbitrary sample, etc).

- The rationale for the study could be made clearer in the introduction. The review of existing literature in this domain is sparse – it is not clear what knowledge gap the study is trying to fill. How does this work build on previous studies and/or extant knowledge in this domain? Why focus on these 11 papers? Why focus on PLOS ONE?

- It would be helpful to define acronyms e.g., what is a “6 ECTS course”?

- This is unclear and perhaps needs rewording: “For problems with implementation specifics for methods described in the papers”

- “RequirementR.3 is important to be able to contact the authors.” – this appears to just be a restatement of the requirement rather than a justification for including it.

- To ensure the reproducibility of their own analyses, the authors may wish to consider saving and sharing the computational environment in which the analyses were performed. Various tools are available to achieve this e.g., Docker, Binder, Code Ocean.

Reviewer #2: What did they do

The authors tried to reproduce 11 statistical analyses published in PLOS ONE that used longitudinal data. This was done by cleverly making use of student labor in the light of a university course. For each paper, a detailed summary file on the OSF describes the study, the model, the analyses, and potential deviations in results. Those files further contain the used R code allowing the verification of this reproducibility study (Personally, I did not make use of that possibility!).

General remarks

I believe this work to be an important contribution to open science and a service to the scientific community in general. The manuscript is well-written and the authors delightfully refrained from being unnecessarily complicated. To put this work into perspective with similar empirical work on reproducibility in psychology, I suggest giving a more detailed description of methods, results, and, implications in the main manuscript. As of now, I am not sure which conclusions to reach about the state of reproducibility in PLOS one. A more detailed summary of the findings is particularly important in this case because each summary was written by a different teams of students making it very time-consuming to extract all the important information.

Major remarks

• I am confused as to the nature of this manuscript. Does it constitute a registered report? If so, the manuscript should clearly indicate what part of the work was done prior to the submission of the registered report and what was done afterward.

• I am missing a (short) Method section where you describe the timeline of the conducted reproductions (when where authors contacted to provide code of analysis?, how did the students work on the assignment?, how (much) assistance did they receive from the teaching team?) In line 340 an internal peer review is mentioned – please provide more information on that.

• I agree with what is being said in lines 208-212, however, I would like to have precise information about when the magnitude of the effect is the same. Further, the possibility of achieving the same numbers by deliberately deviating from the method description of the paper should be discussed as this has implications on the implied interpretations.

• Roughly, we can group reproduction failures into 3 groups: Reporting errors in the paper, insufficient/incorrect description of methods or data that prevent reproduction, and software/algorithm differences. I would like to know for each reproduction failure the group to which it likely belongs. Since you exclusively used R in your reanalyses whereas only 1 of the 11 papers did so, I think it is important to discuss software differences (including differences in algorithms, default/starting values..) in detail. Whereas software differences are negligible for simple designs as ANOVA and t-tests, this cannot just be assumed for GEEs and GLMMs. A discussion of software differences is, for example, important to interpret the results for paper #1 (lines 225 to 234) and also line 257. Looking at your summary file of this paper “essay_01.pdf” it turns out that you have deviated in multiple ways for a multitude of reasons from what was described in the paper. As a result, it is hard to judge whether the original analysis contains reporting errors or not. your analysis of paper #1 A related issue is when you apply a different optimization algorithm. It might be of interest to try to reproduce those papers where the reproduction attempt was unsuccessful (and where the provided data does not seem to be the culprit) via the software package (and the functions therein) used by the respective authors.

• 233 – If you believe that your R code does not converge properly, it should be changed until it does, no? If you are unable to fit the model in R, it cannot be judged whether the published results are approximately correct or not. Now, all we know is that the students assigned to this paper were unable to properly fit the statistical model to the data via R.

• 316 – I would mention that in the abstract. The current abstract might give an incorrect impression as it is nowhere mentioned that the stated results involved author assistance (ideally, reproducibility in an open data journal should be possible without contacting the authors!)

Minor remarks

• 4 – use reproduction instead of replication. More generally, I suggest to use reproduce/reproducibility to describe computational reproducibility and replicated/replicability for new studies involving different data as this terminology is most commonly used in Psychology nowadays.

• 12 - Longitudinal data includes variables that are measured repeatedly over time but those variables do not necessarily have to do with humans.

• 111 – I would choose a more descriptive figure caption.

• 112 – I would refer to R.1 in singular (i.e. requirement R1)

• 130 – Is there the possibility of including additional papers? If so, I would like to see reproduction attempts of the 2 papers were the authors “prohibited” the use of their work to be reproduced.

• 139-143 – Did you try to reproduce ALL figures and numbers reported in the paper that were related to the longitudinal study? If not, what was omitted and why? Please add 1 or 2 clarification sentences.

• 144 – Reproducing someone else’s analysis typically involves many RDFs, yes. But, it does not make sense to say (line 145) that there were many decisions that would adhere to YOUR steps 1-4. Instead, you should write that there are multiple ways to read in/process/analyze the provided data that are not in disagreement with what is stated in the paper or the supplementary material.

• 147 – Please be more specific.

• 153 – The title is not self-explanatory, especially because it is written in the present tense. Maybe “Which statistical methods were used by the papers” instead.

• 162 – “according to statisticians” I would refrain from using such a phrase. Instead, just cite relevant papers arguing for GLMMs over GEEs and, potentially, summarize some of their advantages.

• 172 – See comments for 153 above

• 184 – Did you always ask the authors for their source code. If not, when (before or after the 1st reproduction attempt?) did you ask for it. You provide some information in lines 201 to 207, but, I would like to know the specific time-line, and I want to know what was planned a-priori and what was ad hoc.

• 247 – Where is the search procedure mentioned?

• I would like to see the implications of the non-reproducible findings discussed. How many unreasonable original analyses (& conclusions drawn from it) could be identified? I know that this type of finger-pointing is uncomfortable, especially since only work from authors that provided both their data and responded to your e-mails were included in your sample, yet, it is important to estimate the rate of reporting errors and irreproducible findings in Psychology.

Comments about Review and response to the registered report

• The updated outline of the aim of this study is “Our aim with this study is to better understand the current practices in 11 PLOS ONE papers dealing with longitudinal data in terms of methodology applied but also in how results were computed and how it is made available for readers.” I find this unnecessary complicated and, more importantly, it does not reflect the content of your study well at all. Wasn’t the aim of this study to assess the extent to which papers analyzing longitudinal data in PLOS ONE could be reproduced by independent researchers.

6. PLOS authors have the option to publish the peer review history of their article (what does this mean?). If published, this will include your full peer review and any attached files.

Reviewer #1: No

Reviewer #2: **Yes: **Richard Artner

---

## [Author Response · Author response to Decision Letter 0]

7 Jan 2021

---

output:

 pdf_document: default

---

# PONE-D-20-25993: A computational reproducibility study of PLOS ONE articles featuring longitudinal data analyses

PLOS ONE

Dear Dr. Seibold,

Thank you for submitting your manuscript to PLOS ONE. After careful consideration, we feel that it has merit but does not fully meet PLOS ONE’s publication criteria as it currently stands. Therefore, we invite you to submit a revised version of the manuscript that addresses the points raised during the review process.

The two reviewers and I agree that your study was well done, clearly reported, and relevant, and that your work contributes to our understanding of reproducibility of studies using longitudinal analyses. Yet, several issues need to be dealt with in your revision.

First, the two reviewers requested some additional clarity of the type of submission (registered report or not). I apologize for not indicating to the reviewers that your work was originally submitted as a registered report but later handled as a standard submission. Please clarify this. 

Second, both reviewers asked you to provide additional details of the methods, sampling, operationalizations, and data access. Both reviewers provided detailed feedback on the reporting and analyses that I ask you to consider as you revise your submission.

Third, the reviewers indicate that the main goals and main results could be presented more clearly in several sections of the manuscript. It is also import to be clear on your definition of reproducibility as you present the main results in the abstract. 

We look forward to receiving your revised manuscript.

Kind regards,

Jelte M. Wicherts

Academic Editor

PLOS ONE

#### Journal Requirements:

**Ok.**

- A marked-up copy of your manuscript that highlights changes made to the original version. You should upload this as a separate file labeled 'Revised Manuscript with Track Changes'.The name of the colleague or the details of the professional service that edited your manuscript

**Ok.**

3. In your manuscript text, we note that "We did not choose the papers randomly, but based on the set of potential papers given to us by PLOS ONE and then selected all papers meeting our criteria". In the Methods, please ensure that you have included the following:

- Information about how the initial set of 57 papers was selected, including any inclusion/exclusion criteria applied, so that other interested researchers can reproduce this analysis. We would also recommend that the complete set of 57 initial papers be provided as a supplementary information file.

- The complete inclusion/exclusion criteria used to select the initial set of 14 papers from the 57 papers that were identified.

- The complete inclusion/exclusion criteria used to select the final set of 11 papers.

**Thank you. We updated the manuscript accordingly. Please let us know if we should move any further information from Figure 1 to the text.**

4. Please amend the manuscript submission data (via Edit Submission) to include authors Severin Czerny, Siona Decke, Roman Dieterle, Thomas Eder, Steffen Fohr, Nico Hahn, Rabea Hartmann, Christoph Heindl, Philipp Kopper, Dario Lepke, Verena Loidl, Maximilian Mandl, Sarah Musiol, Jessica Peter, Alexander Piehler, Elio Rojas, Stefanie Schmid, Hannah Schmidt, Melissa Schmoll, Lennart Schneider, Xiao-Yin To, Viet Tran, Antje Volker, Moritz Wagner, Joshua Wagner, Maria Waize, Hannah Wecker, Rui Yang, Simone Zellner.

**Done.**

### 5. Review Comments to the Author

**Thank you so much for your thorough and constructive feedback. We truly think that your input has improved the paper and have rarely seen such helpful reviews. We hope that we have answered all your questions to your satisfaction. Please let us know if we missed or misunderstood anything. All text changes are marked in blue.**

#### Reviewer #1: Manuscript ID: PONE-D-20-25993

Manuscript title: A computational reproducibility study of PLOS ONE articles featuring longitudinal data analyses

##### Summary

This article reports a retrospective observational study designed to test the analytic reproducibility of a small set of PLOS ONE articles containing longitudinal analyses. Specifically, as a part of a class exercise, the authors and their students attempted to repeat the original analyses performed in the selected articles to see if they could obtain similar results. A range of difficulties reproducing the original results were encountered – some of which could be resolved through contact with the original authors. The generalizability of the results is quite limited due to the small sample size and somewhat ad-hoc sampling procedures; however, the authors appropriately calibrate their conclusions to the evidence, for example stating that “We can and should not draw conclusions from our findings on the 11 selected papers on the broader scientific landscape.”

Generally, the paper is clearly written and is concise, though I think some important information is absent (see detailed comments below). The study appears to be transparently reported with analysis scripts and data for each reproducibility attempt made publicly available in an Open Science Framework repository. I only checked this repository superficially – one issue is that I could not seem to identify a data file for the study (see comment below) which needs to be addressed.

Important note: After reading the paper and writing my review, I was surprised to find a document included after the reference section with the title “Review and response to the registered report”. I was not informed that this study was a registered report and as far as I can tell this is not mentioned in the manuscript aside from this appended document. Can the circumstances of this study be clarified? If this is a registered report then I feel I should be given more information. Most importantly, I need to see the original registered protocol. I would also like to see the prior review history and know whether the original stage 1 reviewers are also appraising the stage 2 manuscript.

**We had initially submitted a registered report for this study to PLOS ONE, but since the review process took longer than the project, we decided -- together with PLOS ONE editors -- that it would be more scientifically sound to withdraw the submission and only submit the final paper. We submitted including the full history of reviews and hope you will receive the full information in the upcoming round of reviews.**

##### Major Comments

- The exact operationalization of several of the outcome variables could be made clearer. For example, for the question “Are the methods used clearly documented in paper or supplementary material (e.g. analysis code)?” what was considered ‘clear’ vs. ‘unclear’? For “Do we receive the same (or very similar) numbers in tables, figures and models?” what was considered ‘similar’ vs. ‘dissimilar’? For “What are characteristics of papers which make reproducibility easy/possible or difficult/impossible?” – exactly what characteristics were examined?

**We added a more detailed explanation what we considered a clear description in the methods section. As we only used R in our analysis we speak of a clear documentation, if it allowed us to follow the authors steps in a different software environment. As similar outputs we defined, if they have the same interpretation, as some deviations are expected. If already deviations in simple summary tables occured we immediatly contacted the authors, to get more information on the data. The characteristics examined were based on the discussion of the individual papers and what we identified as the most likely reasons for success/failures. We added a line to point this out.**

- The oversight of the student work could be described in more detail – did teachers fully

**We added two sentences which we hope will clarify the setup further: "Internal peer-reviews ensured that all groups had a solid technical setup. Finally all projects were carefully evaluated by the teachers and updated in case of problems."**

- In the methods section, the sampling procedure is somewhat unclear – for example, “57 papers were initially screened. The PLOS website search function was utilized to scan through PLOS ONE published works. Key words used were “mixed model”, “generalized estimating equations”, “longitudinal study” and “cohort study”. 14 papers fulfilled the criteria and were selected”. Where does the number 57 come from? For the 14 papers – did they fulfill the search criteria? Or the criteria in Fig 1? Or both?

**The phrasing was a bit unfortunate. Thank you for spotting this. We hope it is more clear now.**

- It seems to me that the selection criteria will have heavily biased the results in favour of positive reproducibility outcomes – for example, only studies that had confirmed data availability were selected and only studies where authors replied to contact and were favourable to the reproducibility attempt were included. Because these factors probably influenced the results quite substantially, I’d suggest this bias is mentioned in key parts of the paper like the abstract and introduction.

**Very good point. We included a sentence ("Inclusion criteria were the availability of data and author consent.") in the abstract and another ("Note that we only selected papers which provided data openly 

online and where authors agreed with being included in the study.

We assume that this leads to a positive bias in the sense that other papers 

would be more difficult to reproduce.") in the introduction.**

- I examined the OSF repository (https://osf.io/xqknz/) for this study and it was unclear to me where to find the data for this study (I could find data for the studies that the authors attempted to reproduce). Could clear instructions be provided on how to find and access the study data?

**The data can either be found in the data folder in the respective paper folder (e.g. `papers/01_2017_Wagner/data`) or it is automatically downloaded in the provided code.**

- Could it be clarified if all analyses reported in eligible papers were examined or just the subset pertaining to longitudinal analyses?

**We focused on the longitudinal data analyses. We clarified this in the text.**

- It is reported that for one article partial analysis code was available and for a second article the full analysis code was made available during email exchanges with the original authors. Its not clear whether all original authors were explicitly asked to make their code available – could this be clarified? If so, what were the responses to this query? Were reasonable justifications provided for not sharing the original code?

**We are unsure in how much detail we are allowed to share the email exchanges openly. Aside from the initial contact, email exchanges were not standardized and authors were not asked if we would be allowed to share the details of the conversation. Not all authors were asked to share the code. We hope this answers your questions.**

- The operational definition of reproducibility could be made clearer in the methods section (and perhaps also in the introduction) – in the results section the authors state “we define similar results as having the same implied interpretations” – this seems to be a less strict definition than used in other studies of analytic reproducibility (e.g., Hardwicke et al., 2018; 2020; Minocher et al., 2020). Some clarification and comment on this would be helpful for understanding the results.

**Thank you for the interesting references. We agree that our operationalisation is less strict than others, such as the ones quoted by you. The main reason for this is, that we deliberately deviate from the software used by the authors, so deviation is to be expected to a certain degree and does not necessarily imply any kind of error on the side of the authors. However a detailed enough method description to be able to reach similar results, when using a different software implementation is in our eyes a minimal requirement for successful reporting. We added more explanation to the methods section.**

- I think it would be informative to mention in the abstract how many analyses were reproducible only when assistance was provided by original authors.

**Thank you for the suggestion, we added the number to the abstract.**

- This sentence in the discussion is unclear – “We did not choose the papers randomly, but based on the set of potential papers given to us by PLOS ONE and then selected all papers meeting our criteria (see Figure 1).” If the papers were given to the authors by PLOS ONE then this needs to be mentioned and explained at least in the methods section.

**Thank you for spotting. We added the information in the methods section.**

##### Minor Comments

- Terminology usage in this domain is diverse and sometimes contradictory (see e.g., https://doi.org/10.1126/scitranslmed.aaf5027) – I’d recommend including explicit definitions and avoiding use of loaded terminology if possible. For example, it would be good to have a clear definition of ‘computational reproducibility’ in the opening paragraph. The authors may also want to consider using the term ‘analytic reproducibility’ instead of computational reproducibility. Researchers in this domain have recently started to draw a distinction between the two concepts and the former seems more applicable to what the present study has addressed. The distinction is discussed in this article - https://doi.org/10.31222/osf.io/h35wt – specifically, “Computational reproducibility is often assessed by attempting to re-run original computational code and can therefore fail if original code is unavailable or non-functioning (e.g., Stodden et al., 2018; Obels et al., 2019). By contrast, analytic reproducibility is assessed by attempting to repeat the original analysis procedures, which can involve implementing those procedures in new code if necessary (e.g., Hardwicke et al., 2018; Minocher et al., 2020)”

**Thank you very much for hinting us to the article by Hardwicke et al. This is indeed highly relevant to our study as their setup is similar to ours. Furthermore we now mention the term analytic reproducibility and cite LeBel et al.**

- An additional point on terminology - use of the term ‘replication’ (e.g., in the abstract and introduction) should perhaps be avoided if possible in this context because it is often used to mean “repeating original study methods and obtaining new data” – whereas here it is being used synonymously with computational reproducibility to mean “repeating original study analyses on the original data” (see http://arxiv.org/abs/1802.03311)

**You are completely right. We believe this increases clarity in our manuscript. Thank you!**

- I felt the study design could be made much more explicit in the introduction. For example, “The articles we chose are [1–11]” – briefly mentioning the sampling procedure would be helpful here so the reader can understand the study design (e.g., was it a random sample, arbitrary sample, etc).

**We added the sentence "They are all PLOS ONE papers which fulfilled our selection criteria [...] in March 2019."**

- The rationale for the study could be made clearer in the introduction. The review of existing literature in this domain is sparse – it is not clear what knowledge gap the study is trying to fill. How does this work build on previous studies and/or extant knowledge in this domain? Why focus on these 11 papers? Why focus on PLOS ONE?

**Thank you. This is a very important question which we now answer in the introduction.**

- It would be helpful to define acronyms e.g., what is a “6 ECTS course”?

**Thanks for spotting. ECTS are credit points according to the European Credit Transfer and Accumulation System. We updated the sentence.**

- This is unclear and perhaps needs rewording: “For problems with implementation specifics for methods described in the papers”

**We agree. We did so and hope it is more understandable now: "Students were advised to contact the authors directly in case of unclear specifications of methods."**

- “RequirementR.3 is important to be able to contact the authors.” – this appears to just be a restatement of the requirement rather than a justification for including it.

**Thanks for spotting all these small yet important things. We updated this.**

- To ensure the reproducibility of their own analyses, the authors may wish to consider saving and sharing the computational environment in which the analyses were performed. Various tools are available to achieve this e.g., Docker, Binder, Code Ocean.

**Some of the computations run very long and require server usage which makes it difficult to use the suggested solutions (our University servers do not allow usage of Docker). As we are of course aware of the issue, we decided to provide the `SessionInfo()` for each paper instead. We realized that this information is not provided in the paper and we added it (see Compuational Details).**

#### Reviewer #2: What did they do

The authors tried to reproduce 11 statistical analyses published in PLOS ONE that used longitudinal data. This was done by cleverly making use of student labor in the light of a university course. For each paper, a detailed summary file on the OSF describes the study, the model, the analyses, and potential deviations in results. Those files further contain the used R code allowing the verification of this reproducibility study (Personally, I did not make use of that possibility!).

##### General remarks

I believe this work to be an important contribution to open science and a service to the scientific community in general. The manuscript is well-written and the authors delightfully refrained from being unnecessarily complicated. To put this work into perspective with similar empirical work on reproducibility in psychology, I suggest giving a more detailed description of methods, results, and, implications in the main manuscript. As of now, I am not sure which conclusions to reach about the state of reproducibility in PLOS one. A more detailed summary of the findings is particularly important in this case because each summary was written by a different teams of students making it very time-consuming to extract all the important information.

##### Major remarks

- I am confused as to the nature of this manuscript. Does it constitute a registered report? If so, the manuscript should clearly indicate what part of the work was done prior to the submission of the registered report and what was done afterward.

**We had initially submitted a registered report for this study to PLOS ONE, but since the review process took longer than the project, we decided -- together with PLOS ONE editors -- that it would be more scientifically sound to withdraw the submission and only submit the final paper. We submitted including the full history of reviews and hope you will receive the full information in the upcoming round of reviews.**

- I am missing a (short) Method section where you describe the timeline of the conducted reproductions (when where authors contacted to provide code of analysis?, how did the students work on the assignment?, how (much) assistance did they receive from the teaching team?) In line 340 an internal peer review is mentioned – please provide more information on that.

**Thank you. We incorporated this suggestion.**

- I agree with what is being said in lines 208-212, however, I would like to have precise information about when the magnitude of the effect is the same. Further, the possibility of achieving the same numbers by deliberately deviating from the method description of the paper should be discussed as this has implications on the implied interpretations.

**We agree and added such comments in the discussion of the individual papers, when the difference was not negligible. In general we only deviated from the method description when it was unclear how to proceed, even after contacting the authors. In some cases, we also performed sensitivity analyses and mention the results, but they are not used to judge reproducibility.**

- Roughly, we can group reproduction failures into 3 groups: Reporting errors in the paper, insufficient/incorrect description of methods or data that prevent reproduction, and software/algorithm differences. I would like to know for each reproduction failure the group to which it likely belongs. Since you exclusively used R in your reanalyses whereas only 1 of the 11 papers did so, I think it is important to discuss software differences (including differences in algorithms, default/starting values..) in detail. Whereas software differences are negligible for simple designs as ANOVA and t-tests, this cannot just be assumed for GEEs and GLMMs. A discussion of software differences is, for example, important to interpret the results for paper #1 (lines 225 to 234) and also line 257. Looking at your summary file of this paper “essay_01.pdf” it turns out that you have deviated in multiple ways for a multitude of reasons from what was described in the paper. As a result, it is hard to judge whether the original analysis contains reporting errors or not. your analysis of paper #1 A related issue is when you apply a different optimization algorithm. It might be of interest to try to reproduce those papers where the reproduction attempt was unsuccessful (and where the provided data does not seem to be the culprit) via the software package (and the functions therein) used by the respective authors.

**We agree with this insightful comment. In this study, software/algorithm differences are expected, as we use different software as most of the original studies. Due to this, differences in coefficients, especially in complex models, are expected to occur. We cannot differentiate between reporting errors and errors due to different implementations. The only errors, that we can spot with reasonable confidence are if the methods and data description are not clear enough for us to reach similar results using different implementations. This means, that starting values and other parameters used in the optimization need to be provided, if they can have a high impact on the results. If they were not provided, we relied on the standard settings in the respective R-libraries and reasonable trial and error. So for this study, we can only answer the question "can we reach the same interpretations following the authors descriptions but using different implementations". This is less strict than other operationalisations of reproducibility, but in our opinion reflects a very common situation for researchers. We added a column to the results table, that consists of the most likely reason for failures of reproduction for parts or all results.**

- 233 – If you believe that your R code does not converge properly, it should be changed until it does, no? If you are unable to fit the model in R, it cannot be judged whether the published results are approximately correct or not. Now, all we know is that the students assigned to this paper were unable to properly fit the statistical model to the data via R.

**Thank you for your comment. We did this study under the assumption of having access to only free software. We believe that it is too much to ask from someone aiming to reproduce a paper to write new software. Using licensed products comes with the possibility of using methods that are not openly available at all, which was the case here. We added more information on this in the manuscript.**

- 316 – I would mention that in the abstract. The current abstract might give an incorrect impression as it is nowhere mentioned that the stated results involved author assistance (ideally, reproducibility in an open data journal should be possible without contacting the authors!)

**Thanks for spotting this. We completely agree and added the sentence "For all but two articles we had to contact the authors to reproduce results."**

##### Minor remarks

- 4 – use reproduction instead of replication. More generally, I suggest to use reproduce/reproducibility to describe computational reproducibility and replicated/replicability for new studies involving different data as this terminology is most commonly used in Psychology nowadays.

**You are completely right. We believe this increases clarity in our manuscript. Thank you!**

- 12 - Longitudinal data includes variables that are measured repeatedly over time but those variables do not necessarily have to do with humans.

**Done, thanks.**

- 111 – I would choose a more descriptive figure caption.

**Done, thanks.**

- 112 – I would refer to R.1 in singular (i.e. requirement R1)

**Done, thanks.**

- 130 – Is there the possibility of including additional papers? If so, I would like to see reproduction attempts of the 2 papers were the authors “prohibited” the use of their work to be reproduced.

**We deliberately chose not to work with papers where the authors are not ok with us reproducing the paper as it did not fit in the context of our study. We hope you understand.**

- 139-143 – Did you try to reproduce ALL figures and numbers reported in the paper that were related to the longitudinal study? If not, what was omitted and why? Please add 1 or 2 clarification sentences.

**Thank you for finding this missing information. We added "If many analyses were performed, we focused on the analyses of longitudinal data."**

- 144 – Reproducing someone else’s analysis typically involves many RDFs, yes. But, it does not make sense to say (line 145) that there were many decisions that would adhere to YOUR steps 1-4. Instead, you should write that there are multiple ways to read in/process/analyze the provided data that are not in disagreement with what is stated in the paper or the supplementary material.

**You are absolutely right. We updated the text.**

- 147 – Please be more specific.

**We added more context and hope it is more understandable now.**

- 153 – The title is not self-explanatory, especially because it is written in the present tense. Maybe “Which statistical methods were used by the papers” instead.

**Good point. We updated the caption.**

- 162 – “according to statisticians” I would refrain from using such a phrase. Instead, just cite relevant papers arguing for GLMMs over GEEs and, potentially, summarize some of their advantages.

**Thank you. Now that I read it again, it sounds funny to me, too. We updated the sentence.**

- 172 – See comments for 153 above

**Good point. We updated the caption.**

- 184 – Did you always ask the authors for their source code. If not, when (before or after the 1st reproduction attempt?) did you ask for it. You provide some information in lines 201 to 207, but, I would like to know the specific time-line, and I want to know what was planned a-priori and what was ad hoc.

**We did attempt to do everything on our own. Only if necessary information was not available, we contacted the authors. We are now making this more clear in the text.**

- 247 – Where is the search procedure mentioned?

**We updated the description to make it more clear how the search procedure was conducted: "The PLOS website search function was utilized to scan through PLOS ONE published works. Key words used were ..."**

- I would like to see the implications of the non-reproducible findings discussed. How many unreasonable original analyses (& conclusions drawn from it) could be identified? I know that this type of finger-pointing is uncomfortable, especially since only work from authors that provided both their data and responded to your e-mails were included in your sample, yet, it is important to estimate the rate of reporting errors and irreproducible findings in Psychology.

**Out of the papers, that were not reproducible at all for us, in one case we were unable to preprocess the data, in one the failure was clearly due to implementation differences and only in one case we identified a potentially unreasonable analysis. Of course for the two papers mentioned previously we just cant know, as we were unable to reach results ourselfs.**

- The updated outline of the aim of this study is “Our aim with this study is to better understand the current practices in 11 PLOS ONE papers dealing with longitudinal data in terms of methodology applied but also in how results were computed and how it is made available for readers.” I find this unnecessary complicated and, more importantly, it does not reflect the content of your study well at all. Wasn’t the aim of this study to assess the extent to which papers analyzing longitudinal data in PLOS ONE could be reproduced by independent researchers.

**You are right, thanks. We updated the section.**

---

## [Decision Letter · Decision Letter 1]

9 Mar 2021

PONE-D-20-25993R1

A computational reproducibility study of PLOS ONE articles

featuring longitudinal data analyses

PLOS ONE

Dear Dr. Seibold,

Thank you for revising your manuscript submitted for publication in PLOS ONE. Please accept my apologies for the relatively slow processing of your revision caused by several factors including me being on parental leave around the birth of our daughter and me having to homeschool her three proud brothers during the lockdown .

The remaining reviewer (Reviewer 1 was unfortunately no longer available) and I agree that you responded very well to the issues raised in the earlier round and that your submission is very close to being publishable in PLOS ONE. The reviewer raises some minor issues that can be readily dealt with in the revision or responded to in your letter (in case you choose not to follow the suggestion). Also, please consider adding some references to some relevant recent studies on reproducibility and sharing of syntax and computer code and update the references referring to pre-prints that have appeared in the meantime. If you respond well to the remaining minor issues, I except to make a quick decision on your manuscript without needing to resend it out for review. I am looking forward to seeing your rigorous and interesting work appearing in print.

We look forward to receiving your revised manuscript.

Kind regards,

Jelte M. Wicherts

Academic Editor

PLOS ONE

Journal Requirements:

Reviewers' comments:

Reviewer's Responses to Questions

**Comments to the Author**

1. If the authors have adequately addressed your comments raised in a previous round of review and you feel that this manuscript is now acceptable for publication, you may indicate that here to bypass the “Comments to the Author” section, enter your conflict of interest statement in the “Confidential to Editor” section, and submit your "Accept" recommendation.

Reviewer #2: All comments have been addressed

2. Is the manuscript technically sound, and do the data support the conclusions?

Reviewer #2: Yes

3. Has the statistical analysis been performed appropriately and rigorously? 

Reviewer #2: I Don't Know

4. Have the authors made all data underlying the findings in their manuscript fully available?

Reviewer #2: Yes

5. Is the manuscript presented in an intelligible fashion and written in standard English?

Reviewer #2: Yes

6. Review Comments to the Author

Reviewer #2: I find that this revised paper well-structured and easy to read, and I can recommend its publication as new insights on reproducibility are much-needed. What I would like to see added is a more thorough discussion of the state of reproducibility in PLOS ONE (as well as more generally in psychology) together with its implications (both in the introduction and the discussion section). In particular, this paper should include references to all relevant literature on this topic (see minor remarks below). Otherwise, a reader of this paper will not be made aware of other important empirical findings on this topic.

Below some minor remarks hopefully further improve the quality of this paper.

• Abstract: The last sentence only states quite obvious things. I would prefer to read about non-obvious insights gained in light of this study.

• Line 4: please include important recent studies on reproducibility such as

o Artner, R., Verliefde, T., Steegen, S., Gomes, S., Traets, F., Tuerlinckx, F., & Vanpaemel, W. (2020). The reproducibility of statistical results in psychological research: An investigation using unpublished raw data. Psychological Methods.

o Maassen, E., van Assen, M. A., Nuijten, M. B., Olsson-Collentine, A., & Wicherts, J. M. (2020). Reproducibility of individual effect sizes in meta-analyses in psychology. PloS one, 15(5), e0233107.

o Obels, P., Lakens, D., Coles, N. A., Gottfried, J., & Green, S. A. (2020). Analysis of open data and computational reproducibility in registered reports in psychology. Advances in Methods and Practices in Psychological Science, 3(2), 229-237.

• Line 33: What did other empirical investigations on reproducibility find?

• Lines 160-162: I would not label a lack of knowledge of the exact calculations performed by the authors due to a lack of information provided as RDFs. RDFs are what the original authors had! Maybe you can write: „The description about all these steps was generelly vague (see classification of reported results in Artner et. al, 2020) meaning that there were multiple ways in line with the descriptions in the originial paper. This study, thus, exposed a large amount of Researcher degrees of freedom [23] coupled with a lack in transparancy about in the original studies.“

• I find the style in which the results section is written weird (until line 251). Why not just describe the results with a reference to the respective table. Now we have Tables first and it is not really clear if the text re-iterates the information in the tables or if additional information is provided. Also, why not merge tables 1, 2 and 3? Table 1 alone does not provide enough information to be included in the main text. Maybe the results section can be structured as follows: 1

• Line 239: When was the magnitude considered to be the same? It is important to exactly describe your criteria here to allow the reader to gauge the overall results of your study. Without knowing whether your criterion was lenient or rather strict, it cannot be done as each and every one of us uses individual metrics to gauge similarity.

• Fig 2: General comment - Without knowing the range of parameter values it is hard to interpret differences in original and reproduced results. Why did you choose to report on the estimates of this article. Why not report, for example, on the results of article [1] instead?

• Lines 261-264: Large differences in magnitude should result in different interpretation even in case of equal signs!!!!!

• Line 299: substantial!

• Line 351: Nevertheless we were only able ….

7. PLOS authors have the option to publish the peer review history of their article (what does this mean?). If published, this will include your full peer review and any attached files.

Reviewer #2: **Yes: **Richard Artner

---

## [Author Response · Author response to Decision Letter 1]

9 Apr 2021

## Review Comments to the Author Reviewer #2: 

I find that this revised paper well-structured and easy to read, and I can recommend its publication as new insights on reproducibility are much-needed. What I would like to see added is a more thorough discussion of the state of reproducibility in PLOS ONE (as well as more generally in psychology) together with its implications (both in the introduction and the discussion section). In particular, this paper should include references to all relevant literature on this topic (see minor remarks below). Otherwise, a reader of this paper will not be made aware of other important empirical findings on this topic.

**Thank you so much Mr Artner for your insightful review and constructive feedback. Please see our response to each point below. New changes in the manuscript are marked in green.**

Below some minor remarks hopefully further improve the quality of this paper.

- Abstract: The last sentence only states quite obvious things. I would prefer to read about non-obvious insights gained in light of this study.

**Thanks for this suggestion. As per the *writing in the sciences* course (see https://youtu.be/xmzUQ46YFiE), we added a sentence on the implications of our findings.**

- Line 4: please include important recent studies on reproducibility such as

 - Artner, R., Verliefde, T., Steegen, S., Gomes, S., Traets, F., Tuerlinckx, F., & Vanpaemel, W. (2020). The reproducibility of statistical results in psychological research: An investigation using unpublished raw data. Psychological Methods.

 - Maassen, E., van Assen, M. A., Nuijten, M. B., Olsson-Collentine, A., & Wicherts, J. M. (2020). Reproducibility of individual effect sizes in meta-analyses in psychology. PloS one, 15(5), e0233107.

 - Obels, P., Lakens, D., Coles, N. A., Gottfried, J., & Green, S. A. (2020). Analysis of open data and computational reproducibility in registered reports in psychology. Advances in Methods and Practices in Psychological Science, 3(2), 229-237.

**Thank you so much for sharing these interesting articles with us! I devoured them and gained many insights on what we ourselves could have done differently :). We included all suggested references and made some changes to the manuscript text.**

- Line 33: What did other empirical investigations on reproducibility find?

**I am not sure if that answers the question but we are writing about this in the first paragraph in the introduction. See lines 2-9.**

- Lines 160-162: I would not label a lack of knowledge of the exact calculations performed by the authors due to a lack of information provided as RDFs. RDFs are what the original authors had! Maybe you can write: „The description about all these steps was generelly vague (see classification of reported results in Artner et. al, 2020) meaning that there were multiple ways in line with the descriptions in the originial paper. This study, thus, exposed a large amount of Researcher degrees of freedom [23] coupled with a lack in transparancy about in the original studies.“

**That is a great suggestion and now thinking about it more, I completely agree with you. We changed the text according to your suggestion (with a few tweaks). Thank you!**

- I find the style in which the results section is written weird (until line 251). Why not just describe the results with a reference to the respective table. Now we have Tables first and it is not really clear if the text re-iterates the information in the tables or if additional information is provided. Also, why not merge tables 1, 2 and 3? Table 1 alone does not provide enough information to be included in the main text. Maybe the results section can be structured as follows: 1

**Each paper has its own story and reasons why it was or wasn't reproducible and what the barriers were. With the individual sections we want to show how diverse the papers were and what types of challenges we faced. We now clarify this in the paper. We use 3 tables instead of one because the table would be too big for a PDF otherwise.**

- Line 239: When was the magnitude considered to be the same? It is important to exactly describe your criteria here to allow the reader to gauge the overall results of your study. Without knowing whether your criterion was lenient or rather strict, it cannot be done as each and every one of us uses individual metrics to gauge similarity.

**Very good point. We use a rather lenient definition in the result tables, but describe each result in more detail, to highlight problems or borderline cases. A formal definition is difficult, as the analyzed models are very different and it is hard to define a single criterion, that captures the results adequately. We believe that our lenient definition, together with the descriptive part works well in this smaller scale study, but probably would be impossible for large scale studies on reproducibility.**

- Fig 2: General comment - Without knowing the range of parameter values it is hard to interpret differences in original and reproduced results. Why did you choose to report on the estimates of this article. Why not report, for example, on the results of article [1] instead?

**This is a very good point. The plot shows regression coefficients from the fitted GEE model. We added a clarification to the paper. We chose to report this article as an example for our definition.**

- Lines 261-264: Large differences in magnitude should result in different interpretation even in case of equal signs!!!!!

**We agree that our conclusion was inconsistent in this case and changed the classification of article [1] to be not reproducible. In the other, reproducible, papers the deviations of coefficients were much smaller.**

- Line 299: substantial!

**Thanks for spotting!**

- Line 351: Nevertheless we were only able ….

**Thanks for spotting!**

---

## [Editor Report · Decision Letter 2]

22 Apr 2021

A computational reproducibility study of PLOS ONE articles

featuring longitudinal data analyses

PONE-D-20-25993R2

Dear Dr. Seibold,

We’re pleased to inform you that your manuscript has been judged scientifically suitable for publication and will be formally accepted for publication once it meets all outstanding technical requirements.

Kind regards,

Jelte M. Wicherts

Academic Editor

PLOS ONE

---

## [Editor Report · Acceptance letter]

10 Jun 2021

PONE-D-20-25993R2 

A computational reproducibility study of PLOS ONE articles
featuring longitudinal data analyses 

Dear Dr. Seibold:

I'm pleased to inform you that your manuscript has been deemed suitable for publication in PLOS ONE. Congratulations! Your manuscript is now with our production department. 

Kind regards, 

on behalf of

Dr. Jelte M. Wicherts 

Academic Editor

PLOS ONE